

# Multi-year surface velocities and sea-level rise contribution of the Basin-3 and Basin-2 surges, Austfonna, Svalbard

Thomas Schellenberger[1], Thorben Dunse[1], Andreas Kääb[1], Thomas Vikhamar Schuler[1], Jon Ove Hagen[1] and Carleen H. Reijmer[2]

[1] Department of Geosciences, University of Oslo, P.O. Box 1047, Blindern, 0316 Oslo, Norway
[2] Institute for Marine and Atmospheric Research, Utrecht University, Princetonplein 5, 3584 CC Utrecht, the Netherlands

*Correspondence to*: Thomas Schellenberger (thomas.schellenberger@geo.uio.no)

**Abstract.** Basin-3, the largest outlet basin of the Austfonna ice cap, started to surge in autumn 2012. A maximum velocity of 18.8 m d$^{-1}$ was found in December 2012 / January 2013. Here we present a time series of area wide velocity fields from synthetic aperture radar (SAR) offset tracking and Global Positioning System (GPS) data in the aftermath of the velocity maximum, extending the previously published data from May 2013 to July 2016. We find that terminus velocity slowed down by ~50% until spring 2014, whereas the upper parts of the basin continued to speed-up and reached their maximum only in summer 2014. Until the date of writing (July 2016), Basin-3 maintained high velocity with maxima between 8.9 – 11.4 m d$^{-1}$. Summer speed-ups were superimposed even on the otherwise fast surge motion. The total frontal ablation $A_f$ over the period 19 April 2012 to 26 July 2016 was calculated to 22.2 ± 8.1 Gt (5.2 ± 1.9 Gt yr$^{-1}$) from the ice mass flux $q_{fg}$= 33.2 ± 11.5 Gt (7.8 ± 2.7 Gt yr$^{-1}$) and the terminus mass change $q_t$= 11.0 ± 3.4 Gt (2.6 ± 0.8 Gt yr$^{-1}$). Additional advance of the terminus led to a total sea-level rise equivalent of 31.3 ± 11.2 Gt (7.3 ± 2.6 Gt yr$^{-1}$).

This rate of frontal ablation roughly equals previous estimates of both the calving flux and total mass loss from the entire archipelago, resulting in a doubling of the current ice-mass loss from Svalbard. In vicinity of Basin-3, we also observe a terminus advance and a speed-up of the northern part of Basin-2 starting in autumn 2014, with surface velocity reaching 8.71 m d$^{-1}$ in August 2015. The related ice mass loss of Basin-2 between 20 June 2015 and 26 July 2016 amounts to 0.8 Gt (min: 0.3 Gt, max: 1.6 Gt). Accounting also for the replacement of ocean water, we find a total sea-level rise equivalent of 1.1 Gt (min: 0.5 Gt, max: 2.1 Gt).

## 1 Introduction

Glacier dynamics and surge-type glaciers are still poorly understood and their contribution to future sea level rise is not included in recent projections (Church et al., 2013). Glacier surges were initially defined as periodically reoccurring internally triggered instabilities with prominent speed-ups and often associated with considerable terminus advances (Meier and Post, 1969; Sharp, 1988). The observations of the warm-based Variegated glacier (Kamb et al., 1985) led to the formulation of the hydrological switch model. It explains the onset and termination of a surge by changes in the hydrological system at the glacier bed changes from an efficient conduit system to an inefficient linked cavity system and vice versa. The



second model, the thermal switch model (Clarke, 1976; Murray et al., 1998), illustrates the behavior which was observed e.g. at poly-thermal Bakaninbreen (Murray et al., 1998) and Trapridge glacier (Frappé and Clarke, 2007). Here, the surges started after driving stresses increased due to a gain of mass in the reservoir area which led to a positive feedback mechanism: more heat was generated at the bed, the glacier could flow faster, which then again lead to more strain heating and in the end to a

switch from cold based to warm based conditions at the bed.

About 1% of the world-wide glacier population (approximately 2300 glaciers) is known to have surged or show indirect signs of surging (Jiskoot et al., 2000; Sevestre and Benn, 2015). Svalbard is one of the regions where surges occur regularly and up to 345 surge-type glaciers were identified so far (Sevestre and Benn, 2015). In comparison to other geographical clusters, the quiescence phases of 50 – 500 years and active phases of 3–10 years are long on Svalbard (Dowdeswell et al.,

1991). The typical surge-type glacier on Svalbard flows over a soft bed and is polythermal (Hamilton and Dowdeswell, 1996; Jiskoot et al., 2000). Jiskoot et al. (2000) also recognized that steeper glaciers surge more often, which was however not confirmed by Hamilton and Dowdeswell (1996). Instead Hamilton and Dowdeswell (1996) concluded that the surge probability increases with increasing glacier length.

One of the most recent and prominent examples for surge-type behavior on Svalbard is observed on Austfonna, the largest

ice cap in the Eurasian High Arctic (Fig. 1). Dunse et al. (2015) used ground penetrating radar to map the formation of crevasses along two profiles in the accumulation area of Basin-3 between 2004 and 2012. They observed multiannual stepwise acceleration of GPS stations within a distinct flow unit in the north of Basin-3 since 2008. Satellite synthetic aperture radar data from TerraSAR-X (TSX) revealed the expansion of the fast flowing part of Basin-3 in 2012. A hydro-thermodynamic feedback mechanism, which unifies cryo-hydrological warming (Phillips et al., 2010) and hydraulic-

lubrication (Schoof, 2010) and promoted the dynamic instability of the marine-terminating ice cap, was proposed to explain this surge evolution (Dunse et al., 2015). During subsequent summers, meltwater not only lubricated the bed, evident in pronounced speed-ups, but also facilitated successive onset of basal motion in initially slow-flowing, cold-based ice regions, culminating in a massive surge over an expanding area and resulting in substantial calving loss as well as terminus advance.

Similar observations were made on Stonebreen, Edgeøya, where summer velocity also gradually rose over multiple years

since 2011 (Strozzi et al., 2016). Since then, Stonebreen also advanced by ~500 m after a long period of retreat. Increased slope, reduction in ice thickness, surface melt, eventually higher ocean temperatures or combinations of thereof are possible reasons for this surge (Strozzi et al., 2016). The roughly synchronous recent events of Basin-3, Basin-2 and Stonebreen suggest that also overarching external factors can play an important role in the initiation of surges.

In this study we monitor the surface velocity of Basin-3 throughout the 5 years of its active surge with special emphasis on

the evolution of the surge after the main velocity peak in early 2013. We also extend the frontal ablation estimate and sea level rise contribution of Basin-3 for the period 19 April 2012 – 9 May 2013 of Dunse et al. (2015) until 26 July 2016. Another novelty is the examination of the speed-up of the northern part of Basin-2 which started in autumn 2014 and the calculation of its frontal ablation and sea level rise contribution for the period 20 June 2015 – 26 July 2016.



## 2 Study area

The Austfonna ice cap covers most of the Nordaustlandet Island (Fig. 1a and 1b) in NE-Svalbard and is with its 7800 km², the largest ice cap in the Eurasian High Arctic. At the main dome the elevation is ~800 m (Moholdt and Kääb, 2012) and the ice is approximately 600 m thick (Dowdeswell et al., 1986). The ice cap has a polythermal bed, i.e. it is warm-based in the
interior (Zagorodnov et al., 1989) and cold-based along the margins, with the only exceptions of a few fast moving outlets driven by basal motion. The ice cap can be divided in several basins, from which the ones to the south-east (including Basin-3 and Basin-2) are terminating as continuous calving front in the Barents Sea.

Basin-3 (Fig 1c) is the largest basin and has an area of approximately 1200 km² and a length of 60 km. Prior to the surge, distinct fast flow was topographically constrained by Isdomen, a bedrock outcrop to the north, and stagnant ice to the south,
including deep parts of the subglacial valley. Today the ice flow follows a subglacial valley, from which the lower third is grounded below sea level. The basin is in large parts underlain by marine sediments and is non-floating (Dowdeswell et al., 2008;Solheim, 1991). The ice flow changed rapidly over the past years. In the mid-1990s the northern arm of the basin was flowing at a rate of 200 m yr⁻¹ (Dowdeswell et al., 2008). GPS data available after 2008 revealed a stepwise acceleration until autumn 2012, when the whole basin started to surge (Dunse et al., 2015). A maximum velocity of 18.8 m d⁻¹ was
measured between 28 December 2012 and 8 January 2013 (Dunse et al., 2015).

Basin-2 is a smaller basin to the south of Basin-3. According to Nuth et al. (2013) it has a length of 17 km, a width of 13.5 km and covers 153 km² (Fig 1c). Recent satellite data and velocity maps indicate that it consists of two distinct flow units which are separated by 2.8 km at the front. The northern part which is currently surging was ~8 km wide and reached 13.4 km upglacier in July 2016. The southern part reaches further south into Bråsvellbreen as compared to the outlines in 2001
from Nuth et al. (2013). There are no records of previous surges of Basin-2 (Lefauconnier and Hagen, 1991).

## 3. Methods

### 3.1 Synthetic Aperture Radar offset and speckle tracking

In this study, we closely follow the processing steps of Schellenberger et al. (2015) and Dunse et al. (2015) to derive glacier surface velocity of Basin-3 and Basin-2 by means of SAR offset and speckle tracking. The time-series of velocity maps from
TSX StripMap Mode data (TSX) between 19 April 2012 and 23 November 2013 from Dunse et al. (2015) was prolonged until 23 July 2014. Furthermore a time series from Radarsat-2 Fine Mode data (RS-2 F) was acquired between 19 July 2014 and 29 December 2015. Additionally, Radarsat-2 Wide Fine Mode data (RS-2 WF) were acquired between November 2015 and July 2016 over almost the entire Austfonna ice cap including Basin-3 and Basin-2. Radarsat-2 Wide Mode data (RS-2 W) covering Austfonna and Vestfonna were acquired between 7 May 2013 and 15 November 2013 and related velocity
maps were used to fill the gaps from TSX acquisitions in Fig. 3 and 5. Data specifications and processing parameters are listed in Table 1.





Displacements were derived by using cross-correlation between two consecutive acquisitions (Strozzi et al., 2002). The size of the matching window was adjusted according to the expected maximum displacements during the repeat pass cycle. For the 11 day repeat cycle of TSX (~2m ground resolution) we used a 300 x 344 pixel search window in range and azimuth direction. When one or more acquisition were missing (22 days or longer between two acquisitions) we used a search

window of 599 x 688 pixel. For TSX, the step size between to displacement measurements was set to 50 x 57 pixel in range and azimuth, respectively, to achieve a resolution of the displacement map of ~100 x 100 m. Different matching window sizes and step sizes were used in case of RS-2 F, as the original pixel resolution is coarser (~8 x 5 m). The search window was set to 102 x 162 pixel, the step size to 6 x 10 pixel, leading to a 50 m resolution of the velocity map. For the RS-2 WF data with a similar pixel resolution than RS-2 F, we reduced the search area window to 83 x 152 pixel as we expected

smaller displacements. The step size was set to 10 x 19 pixel (~100 x 100 m displacement map) to reduce processing time for the large scenes covering all of Austfonna.

All velocity maps were then geocoded using a digital elevation model (DEM) of Nordaustlandet (Moholdt and Kääb, 2012). The velocity maps were filtered using MATLAB with a maximum velocity filter, which removes all velocity values higher than a manually extracted maximum value (Table A1 in Appendix A). Afterwards, a standard deviation filter was applied

which removes outliers when the standard deviation in a surrounding 3 x 3 pixel window exceeds the value of fifteen meters per repetition cycle. For the estimation of the ice mass flux of Basin-2 and Basin-3 obvious erroneous velocity estimates remaining in the vicinity of a fixed fluxgate were manually removed and the maps interpolated using inverse distance weighting to provide continuous velocity profiles along the fluxgate.

In case of Basin-3, there were four velocity maps from which the velocity along the fluxgate could only be derived partially

due to the long interval between the acquisitions. We used different methods to fill the remaining gaps. The velocity along the fluxgate at the time steps $t_3$ and $t_6$ (Table A1 in Appendix A) were derived by scaling the remaining estimates as described in Dunse et al. (2015).

The time steps $t_3$, $t_{20}$ and $t_{22}$ were interpolated with $t_i = f \cdot t_{i-1}$. This factor f was derived by scaling the good matches of time steps $t_3$, $t_{20}$ and $t_{22}$ to the velocities of the previous repetition cycles $t_2$, $t_{19}$ and $t_{21}$, respectively, i.e. assuming that the shape of

the velocity profile did not change significantly. This was not the case for period $t_6$, when previously stagnant ice regions started to accelerate. This velocity profile is estimated as the mean of $t_5$ and $t_7$ and evaluated by comparing the resulting profile with the available measurements. Furthermore, $t_{27}$ was scaled from eleven days to seven days (12 July 2014 – 19 July 2014) as the velocity maps from TSX (12 July 2014 – 23 July 2014) and RS-2 (19 July 2014 – 12 August 2014) were overlapping.

For Basin-2, we found larger gaps in the velocity maps, especially along the shear margins. For the calculation of the ice mass flux, the velocity maps interpolated with the IDW algorithm were therefore used as "best guesses". The lower boundary of the ice mass flux estimate was calculated from velocity maps in which the gaps were filled with a fixed value of 0.5 m d$^{-1}$, approximately the minimum velocity occurring at the shear margins. The upper boundary was calculated by filling the gaps with the maximum velocity occurring along the fluxgates. Therefore the minimum and maximum values for ice



mass flux, frontal ablation and sea-level contribution should be interpreted as lower and upper boundaries, rather than strict error estimates.

## 3.2 Velocity time series from Global Positioning System

The Global Positioning System (GPS) velocity time series starting in 2008 from Dunse et al. (2012) and Dunse et al. (2015) was prolonged until July 2016. Originally 'five GPS receivers were deployed along the mid-1990s central flowline, 5 to 21 km upglacier from the calving front (Dunse et al., 2012). We used GPS single-frequency code receivers (L1 band, C/A code only). Geographical positions were logged at hourly intervals, and every third hour for instruments installed after May 2011, at an accuracy typically better than 2 m (den Ouden et al., 2010). Filtering in the time domain was applied to reduce random errors, i.e. a 7-day running mean was applied to the daily mean position, velocities were computed, and, finally, the velocity was smoothed by applying another 7-day running mean (Dunse et al., 2012).' (Dunse et al., 2015). In autumn 2011, GPS #3 stopped recording and in January 2015 GPS #1 and #5 followed.

## 3.3 Accuracy assessment

We compared the GPS velocity to SAR based velocities. For the TSX velocities and GPS velocities we find a fit of $v_{TSX}$ = 0.95 $v_{GPS}$ - 0.03 and a $R^2$= 0.94 (Fig. 2 a). For the RS-2 F data there is a fit of $v_{GPS}$ = 1.02 $v_{RS\text{-}2\,F}$ - 0.10 and an even slightly better $R^2$= 0.98 (Fig. 2 b). Also the RS-2 WF matched well with the GPS data: $v_{GPS}$ = 1.04 $v_{RS\text{-}2\,WF}$ - 0.34; $R^2$ = 0.99 (Fig. 2 c). The standard deviation (SD) is 0.43 for TSX, 0.29 for RS-2 F and 0.13 for RS-2 WF data. Therefore we took the more conservative estimate ($SD_{TSX}$ = 0.43 from TSX) as error in the calculation of the ice mass flux of Basin-3. It is also slightly higher than the error estimate for TSX in Dunse et al. (2015) of 0.37 m $d^{-1}$. As only RS-2 data is used in the calculation of the ice mass flux of Basin-2 we use the $SD_{RS\text{-}2\,F}$ = 0.29 as error estimate.

## 3.4 Frontal ablation

Three individual processes cause frontal ablation $A_f$, namely iceberg calving D, subaerial frontal melting and sublimation $A_{f\,(air)}$, and subaqueous melt $A_{f(water)}$ (Cogley et al., 2011).

As with the velocity maps, we closely follow the approaches used in Schellenberger et al. (2015) and Dunse et al. (2015) to estimate the frontal ablation $A_f$. This procedure can be divided in two steps: the calculation of the ice mass flux through a fluxgate $q_{fg}$ and the glacier mass changes at the terminus $q_t$ due to advance or retreat. $A_f$ is the difference of both:

$A_f = q_{fg} - q_t$

$q_{fg}$ is the ice mass flux through a defined fluxgate, with

$q_{fg} = v_{fg} \cdot H_{fg} \cdot w_{fg} \cdot \rho_{ice}$,

where $v_{fg}$ is the velocity, $H_{fg} = zs_{fg} - zb_{fg}$ is the height of the calving front (with $zs_{fg}$ = surface elevation and $zb_{fg}$ = bedrock elevation), $w_{fg}$ is the width of the terminus and $\rho_{ice}$ is the density of the ice (917 kg $m^{-3}$). As the velocity is still around 10 m $d^{-1}$ we assume basal motion governing total displacement (i.e. depth averaged velocity equals surface velocity).





For the calculation of the frontal ablation it is necessary to derive the areal change $\Delta a_t$ of the glacier between two consecutive acquisitions. Therefore calving front outlines were digitized from geocoded backscatter images. The terminus mass change $q_t$ is then calculated to:

$$q_t = H_t \cdot \frac{\Delta a_t}{\Delta t} \cdot \rho_{ice}$$

with $H_t = zs_t - zb_t$ being the ice thickness derived from surface elevation $zs_t$ and bedrock elevation $zb_t$. The input variables, their sources and uncertainties are detailed in Table A2 for Basin-3 and Table A3 for Basin-2.

### 3.5. Automatic weather station data

Since May 2004, we have operated an Automatic Weather Station (AWS) on Etonbreen, a flow unit of Austfonna draining westwards into Wahlenbergfjorden (Fig. 1, Schuler et al. (2014)). The location (370 m a.s.l.) is close to the long-term
equilibrium line altitude but interannual variations are large (Schuler et al., 2007; Dunse et al., 2009). The station measures and records meteorological and glaciological variables, necessary to assess the surface energy balance (Østby et al., 2013), such as components of the radiation budget, air temperature and humidity and wind speed and direction. In addition, we recorded the vertical distribution of temperature in the seasonal snowpack and in near-surface ice to account for retention of meltwater by refreezing. A more detailed description and quality assessment of the data is found in Schuler et al. (2014).
From the temperature record, we determined positive degree days (PDD), i.e. daily mean temperature above the melting point 0 °C and cumulative positive degree days (CPDD) to semi-quantitatively characterize daily and seasonal meltwater production, respectively.

## 4 Results

### 4.1. Basin-3

#### 4.1.1 Velocity evolution

After the multiannual stepwise acceleration of the northern branch, the acceleration of southern part and the merge of the both fast flowing regions in autumn 2012, the surge reached its maximum in December 2012 / January 2013 with velocities at the front of 18.8 m d$^{-1}$ (Dunse et al. (2015), Fig. 3). Afterwards, velocity at the terminus decreased (see also GPS #1 and #2 in Fig. 4), while further upglacier it kept accelerating (Fig. 3). GPSes #4 and #5 were drawn towards south-east into the
faster flowing part of the stream and therefore show acceleration even after the terminus velocity peak in January 2013 (Fig. 4). GPSes #1 and #2 instead were pushed towards the margins of the area affected by the surge.
In July and August 2013 the summer maximum overlies the surge (Fig. 3 - 5). After that, we found decrease of the velocity until July 2014 at the front and of GPSes #1, #2 and #4. The summer speed-up 2014 started at the beginning of July and the velocity was with ~9-10 m d$^{-1}$ almost constant along the whole flowline profile (Fig. 3 and Fig. 6). Afterwards the surface
velocity is relatively stable with maximum velocities between 8.9 and 11.4 m d$^{-1}$.




### 4.1.2. Frontal ablation and sea-level contribution

Between 19 April 2012 and 26 July 2016 the total ice mass flux through the fluxgate amounted to $q_{fg}$ = 33.2 ± 11.5 Gt (7.8 ± 2.7 Gt $yr^{-1}$) (Tab. 2). The total terminus change $q_t$ in the observation period was 11.0 ± 3.4 Gt (2.6 ± 0.8 Gt $yr^{-1}$). This results in a total mass loss of Basin-3 $q_{mb}$ = 22.2 ± 8.1 Gt (5.2 ± 1.9 Gt $yr^{-1}$). Additionally the advancing terminus replaced $q_{tsw}$ = 9.2 ± 3.2 Gt (2.1 ± 0.7 Gt $yr^{-1}$) of ocean water, leading to a total sea level rise equivalent $q_{sl}$ = 31.3 ± 11.2 Gt (7.3 ± 2.6 Gt $yr^{-1}$). For the ice mass flux, we see an increase from 1.8 Gt $yr^{-1}$ in April 2012 to 13.1 Gt $yr^{-1}$ in January 2013 (Fig. 7). From then on it continuously decreases to 7.7 Gt $yr^{-1}$ until February 2014. The only exception is in August 2013, when it rises to 11.7 Gt $yr^{-1}$ compared to the long repetition cycle before (May-August 2013; 10.4 Gt $yr^{-1}$), due to increased summer velocity. Since February 2014 the ice mass flux fluctuates between 5.8 Gt $yr^{-1}$ and 9.4 Gt $yr^{-1}$ and amounted to 7.8 Gt yr-1 during the last repetition cycle for which we calculated frontal ablation for (2 – 26 July 2016).

## 4.2 Basin-2

### 4.2.1 Speed evolution

Next to Basin-3, neighbouring Basin-2 to the south-west showed a recent speed-up of its northern part (Fig. 6d). We monitored the velocity evolution since 2012 and found a first acceleration in August 2013 to 0.6 m $d^{-1}$ (Fig. 10). It gradually slowed down in winter 2013/2014. Then the velocity at the terminus reached 1 m $d^{-1}$ in July 2014, 2 m $d^{-1}$ in September 2014 and almost 3 m $d^{-1}$ in October 2014. There is a gap in the velocity data in winter 2014/2015 until March 2015. Between mid-March and beginning of May a velocity of 5.3 m $d^{-1}$ was measured 2 km upglacier of the terminus position at that time. The surge also reached regions upglacier with velocity of 3.3 m $d^{-1}$ ca. 4.7 km away from the calving front. The velocity considerably rose in July 2015, coinciding with the extended availability of meltwater percolating to the bed, and peaked at a maximum velocity of 8.7 m $d^{-1}$ in August 2015. The speed-up reached up to more than 7 km upglacier in autumn 2015. From September 2015 on, the velocity gradually slowed down and a velocity of 6.5 m $d^{-1}$ was reached in winter 2015/2016. In July 2016, we observed a speed-up to 8.2 m $d^{-1}$ not reaching the same level as the year before.

The calving front stayed stable between April 2012 and September 2014. Then the speed-up in autumn 2014 coincides with the advance of the calving front. From then on, the glacier advanced by 2.1 km until July 2016.

The southern part also short enormous summer speed-ups in the last years with maximum velocities higher than 5 m $d^{-1}$ in 2015 but always dropped back to close to zero velocities during the winters.

### 4.2.2. Frontal ablation and sea-level contribution

The quality of the RS-2 velocity maps allowed the calculation of the frontal ablation only from June 2015 onwards. We estimated the ice mass flux of Basin-2 for the period 20 June 2015 to 26 July 2016 to $q_{fg}$ = 1.3 Gt (min: 0.7 Gt, max: 2.2 Gt) (Tab.3). The total terminus change $q_t$ in the observation period was 0.4 Gt (min: 0.3 Gt, max: 0.6 Gt). This results in a total



mass loss $q_{mb}$ of 0.8 Gt (min: 0.3 Gt, max: 1.6 Gt). Additionally the terminus replaced $q_{tsw}$ = 0.3 Gt (min: 0.1 Gt, max: 0.5 Gt) of ocean water, leading to a total sea level rise equivalent of 1.1 Gt (min: 0.5 Gt, max: 2.1 Gt).

## 5. Discussion

### 5.1 Basin-3

Dunse et al. (2015) described the evolution of the surge in three phases. In Phase I, a spatially confined fast flowing area is initiated. In Phase 2, we observe the mobilization of the reservoir and the multiannual, stepwise acceleration. Finally, the marginal ice plug is destabilized in Phase 3, which results in the basin-wide surge.

In this study, we monitor the surface velocity of Basin-3 in Phase 4 after the main peak. We find that the surface velocity close to the calving front slows down after January 2013 until the summer peak in July 2013. This is not directly visible in
the SAR data due to the long data gap between May and August 2013 but instead confirmed by the GPS data. Maximum surface velocity is then relatively stable after November 2013 with values between 8.9 and 12.3 m d$^{-1}$.

The SAR and the GPS data also show that the velocity further upglacier still rises after the main peak in velocity. Maximum velocity moves from the calving front to a position about 20 km upglacier. We find that the highest velocity (~10.3 m d$^{-1}$) there is not reached until the summer speed-up in 2014 and therefore 1.5 years after the velocity peak at the terminus (Fig.
3). This flow pattern with two fast flowing regions along the flow line of Basin-3 (Fig. 6c, d) is related to the bedrock topography. The subglacial valley descends towards the deepest part, a trough about 10-17 km upglacier from the terminus. Here the subglacial valley is confined and flow velocity increases through this bottleneck to maintain ice flux. Further downglacier the valley broadens and the glacier flows slower before it accelerates again towards the front.

The surface velocity variations during the summers are related to availability of water at the glacier bed, as we still find
summer maxima in 2014, 2015 and 2016 coinciding with the melt period (Fig. 4). While before the main peak of the surge the summer peak is higher for faster moving areas as seen from GPS data, the magnitude of the summer peaks are more similar between all GPSes in 2014, 2015 and 2016, indicating plug flow. Winter velocities in 2014, 2015 and 2016 are approximately as high as in the winter before the basin-wide surge (January 2012). An extraordinary warm year was 2015 when the CPPDs rose to 572 °C d. This had also influence on the velocity of Basin-3, where autumn velocities stayed higher
than in 2014. From January 2016 on, the basin slowed down until the summer speed-up in July 2016, which was higher than in the two previous years.

The advance and subsequent activation of slow moving areas upglacier, also led to an average surface lowering estimated to 5.5 ± 0.8 m yr$^{-1}$ between 2012 and 2014 (McMillan et al., 2014). This lowering is expressed in a very distinct southern shear margin, as seen in the RS-2 image (Fig. 1a), the Landsat 8 Operational Land Imager image (Fig. 1c), and the velocity maps
(Fig. 6).

The updated frontal ablation estimate highlights that the basin still loses mass at a very high rate after the main velocity peak in winter 2012 / 2013. The total mass loss rate through frontal ablation was calculated to 5.2 ± 1.9 Gt yr$^{-1}$ for the period 19



April 2012 to 26 July 2016. Accounting also for the advance of Basin-3, we found a contribution to sea level rise of 7.3 ± 2.6 Gt yr$^{-1}$ in the period. To place this into perspective, we compare these numbers to results on frontal ablation and surface mass balance estimates for Austfonna and Svalbard. Dowdeswell et al. (2008) estimated the frontal ablation of the whole Austfonna ice cap to about 2.5 ± 0.5 km$^3$ yr$^{-1}$ water equivalent. According to Blaszczyk et al. (2009) the mass loss rate due to

frontal ablation of the whole archipelago is 6.75 ± 1.7 km$^3$ yr$^{-1}$ water equivalent without accounting for surges. Looking at the geodetic mass balance of all glaciers in Svalbard for the period 2003–2008, Moholdt et al. (2010) found a loss rate of 4.3 ± 1.4 Gt yr$^{-1}$, excluding calving front retreat or advance, and 6.6 ± 2.6 Gt y$^{-1}$, when included. These comparisons reveal the importance of the Basin-3 surge for the regional mass balance of Svalbard, leading to a doubling of the current ice mass loss from the entire archipelago.

**5.2 Basin-2**

In 2013 the northern part of Basin-2 already showed a slight increase in velocity, however the front stayed stable. In 2014, the glacier did not slow down after the summer peak but velocity increased instead steadily until summer 2015, coinciding with a significant advance of the terminus. This suggests that similar processes than in Phase 3 of the Basin-3 surge took place, when the frontal ice plug of the southern part of Basin-3 was destabilized and led to a basin-wide surge (Dunse et al.,

2015). The formerly cold-based and stagnant glacier margins were weakened by meltwater which led to enhanced basal lubrication (Schoof, 2010) and cryo-hydrologic warming (Phillips et al., 2010). This led to sheer tearing of the ice-sediment interface at the glacier base (Dunse et al., 2015) and subsequent fast moving by basal motion.

RS-2 offset tracking results of Basin-2 between autumn 2014 and spring 2015 were not good enough to extract velocity along the fluxgate and the flowline, most likely due to too extreme surface feature changes in the course of speed-up and

advance. We also tried to derive ice-surface velocity from Sentinel-1A Interferometric Wide Swath Mode data for this period but were not successful either. The Sentinel-1A satellite has the advantage of a shorter repetition cycle of 12 days compared to 24 days for RS-2. Sparse reliable RS-2 offset tracking matches indicate a continuous speed-up of Basin-2 (Fig. 10). This lack of data is also the reason that the frontal ablation estimate of Basin-2 is only possible from June 2015 on.

One might ask whether it is possible that both surges will merge to one fast flowing unit, similar to what happened with the

northern and southern branch of Basin-3 as the presently fast-flowing parts both basins are separated by only ~1.5 km at their termini. This separation turns out to be due to the bedrock topography. While both branches of Basin-3 flowed in the same valley, the bedrock map indicates a ridge between Basin-2 and Basin-3, which so far prevented both streams from merging.

Also the evolution of the speed-ups of Basin-2 and Basin-3 is comparable to the one observed on Stonebreen, Edgeøya, since 2011 (Strozzi et al., 2016). There, the summer velocity peak grew over multiple years up to ~6.8 m d$^{-1}$ while winter

velocities did not increase stepwise but still varied on an elevated level each year. The speed-up was accompanied by an advance of ~500 m after steady retreat. Although there is a difference in the shape of the three surges, which is likely related to the geometry of the glaciers and their beds, it is well possible that similar mechanisms took place. Altogether, these recent observations of Basin-2, Basin-3 and Stonebreen indicate thus that also external factors can play an important role in the




initiation of such kind of surges. Whether their growing number is a consequence of global warming, which is especially evident in the high arctic or increasingly available remote sensing data is not yet clarified.

## 6. Conclusions and outlook

Basin-3, the largest basin of the Austfonna ice cap is still surging as of July 2016. After the absolute velocity maximum of 18.8 m d$^{-1}$ in December 2012 / January 2013 the glacier started slowing down at the calving front but fast flow expanded upglacier. The highest velocity ~20 km from the terminus was observed during the summer speed-up in 2014. After November 2013 the maximum velocities are relatively constant between 8.9 and 11.4 m d$^{-1}$. In July 2016 the maximum velocity was 10.2 m d$^{-1}$ some 22 km upglacier of the calving front, while at the calving front itself we found 9.9 m d$^{-1}$ during the same period. Short-term speed-ups are observed in each summer after the main velocity peak due to increased melt water availability. We find that the onsets of the summer speed-ups are in phase with CPDDs which suggests a meltwater control We find that these seasonal variations are mainly due to differences in the amount of meltwater reaching the bed.

In the period 19 April 2012 to 26 July 2016 Basin-3 lost 5.2 ± 1.9 Gt yr$^{-1}$ through frontal ablation. Accounting also for the advance of Basin-3 we found a contribution to sea level rise of 7.3 ± 2.6 Gt yr$^{-1}$. Between 20 June 2015 and 26 July 2016 the ice mass loss at the terminus of Basin-2 amounted to 0.7 Gt yr$^{-1}$ (min: 0.3 / max: 1.4 Gt yr$^{-1}$). The sea level rise contribution was estimated to 1.0 Gt yr$^{-1}$ (min: 0.4 / max: 1.9 Gt yr$^{-1}$).

The surge of Basin-2 is after those of Basin-3 and Stonebreen the third one in a row for which available observations strongly suggests external forcing in triggering the instability. We believe that also here the hydro-thermodynamic feedback mechanism to summer melt which triggered the Basin-3 surge plays an important role.

As the typical active surge phase of Svalbard glaciers is 3 to 10 years (Dowdeswell et al., 1991) and with the large accumulation area of Basin-3, we expect the surge to still last for a few years. Therefore further work should include the prolongation of the time series with RS-2 WF data and Sentinel-1 A/B data. The Sentinel-1 A satellite has a repetition cycle of twelve days, which is even shortened to six days when combined with the twin satellite Sentinel-1 B. With images acquired every six days it might be even possible that the coherence is preserved and interferometric approaches are feasible. This would especially allow a more correct estimation of the velocity of the slow flowing parts and a new delineation of Basin-3, which was not possible with the available SAR data until now.



**Author contribution**

T. S. provided Radarsat-2 data, processed TSX and RS-2 data, calculated glacier surface velocities and digitized glacier terminus positions. T.S. also wrote the manuscript and designed the figures. T.D. processed the GPS data, wrote the scripts to calculate the frontal ablation to compare SAR- and GPS-derived velocity and to plot the related figures. He also provided

Fig. 4. T.D., J.O.H. and T.V.S. set up the field experiment. T.V.S. analysed AWS data and gave access to a quality-controlled data set. A.K. provided Radarsat-2 and TerraSAR-X data. C.H.R. provided GPS instruments and access to a quality-controlled data set. All co-authors assisted in data interpretation and commented on the paper.

**Acknowledgements**

T.S. was funded by the Research Council of Norway (RASTAR, 208013), the Norwegian Space Centre as part of European

Space Agency's PRODEX program (C4000106033), and the European Union FP7 ERC project ICEMASS (320816). T.D. was supported by the NordForsk-funded project 'Green Growth Based on Marine Resources: Ecological and Sociological Economic Constraints' (GreenMAR, 61582, Nordic Top-level Research Initiative (TRI)) and the CRYOVEX project (ESA PRODEX C4000110725). A.K also received funding from the European Union FP7 ERC project ICEMASS (320816) and the ESA project Glaciers_cci (4000109873/14/I-NB). Additional support was received from the NordForsk-funded project

'Stability and Variations of Arctic Land Ice '(SVALI, 2011-2016, 24420, TRI).

We would like to thank T. Eiken and G. Moholdt for assistance in the field. The TerraSAR-X data were provided by the German Aerospace Center DLR (LAN_0211). Radarsat-2 Fine Mode data were provided by Canadian Space Agency (CSA) within the SOAR-ESA project 16804. Radarsat-2 Wide Fine Mode data were provided by NSC/KSAT under the Norwegian-Canadian Radarsat agreements 2015-2016 and by the Canadian Space Agency (CSA) within the SOAR-ASI project

ALARM (2925/5231). Landsat-8 OLI data, courtesy of the U.S. Geological Survey, was downloaded from EarthExplorer. The GPSes were funded by the Netherlands Polar Programme (NPP) and the Netherlands Organisation for Scientific Research, Earth and Life Sciences section (NWO/ALW). The GPS were developed and technically supported by W. Boot from the Institute for Marine and Atmospheric Research, Utrecht University, the Netherlands.

**Competing interests**

25  A.K. is a member of the editorial board of the journal.





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



**Figure 1: Study area Nordaustlandet with Basin-3 and Basin-2. (a) The Austfonna ice cap's location within the Svalbard archipelago (inset). Outlines of drainage basins in 2001 (from Nuth et al. (2013), Basin-2: red, Basin-3: green, others: solid grey) and surface elevation contours (50 m interval, solid black) overlain on a TerraSAR-X intensity image of 30 April 2012. Positions of five GPS receivers, the stake, the automatic weather station (AWS) and GPR profiles are marked as well (modified from Dunse et al. (2015)). (b) Bedrock contours (black, colour-filled) are at 25m intervals, with the bedrock sea-level contour highlighted in red and 50m surface elevation contours (white) superimposed (from from Dunse et al. (2015)). (c) Close-up of Radarsat-2 intensity**



image of Basin-3 and Basin-2 of 26 July 2016. Glacier outlines of both basins on 26 July 2016 (coloured) and 2001 (grey, from Nuth et al. (2013)). Basin-2 today has two distinct flow units, North (N) and South (S). Fluxgates and flowlines used to extract velocity profiles for Figures 3, 5, 11, as well as locations of GPS stations on 1 January 2015 are also shown.

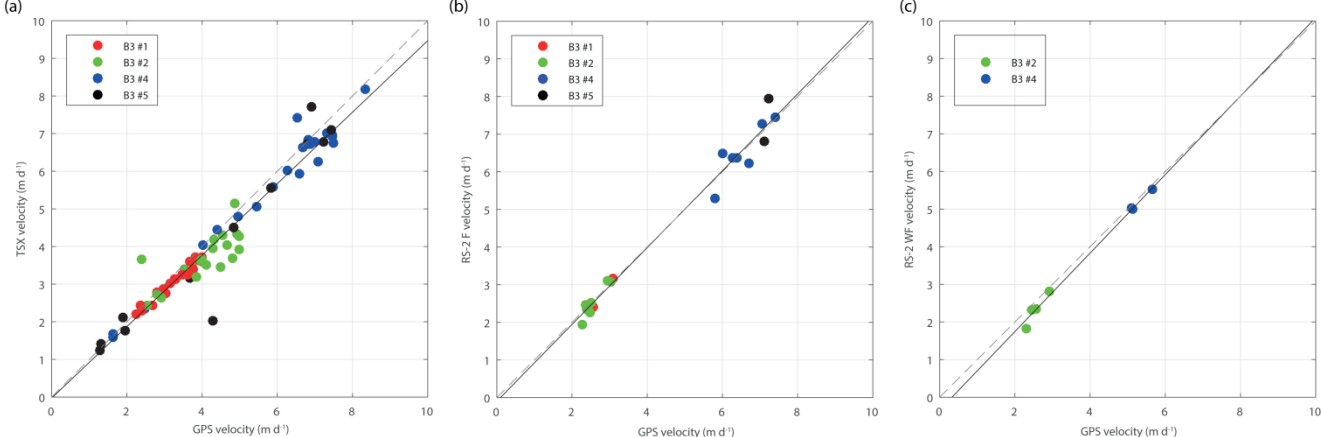

5   Figure 2: Validation of glacier velocity: velocity extracted from SAR maps at the position of GPSes and plotted against GPS velocity (a) TSX vs. GPS (b) RS-2 F vs. GPS (c) RS-2 WF vs. GPS.

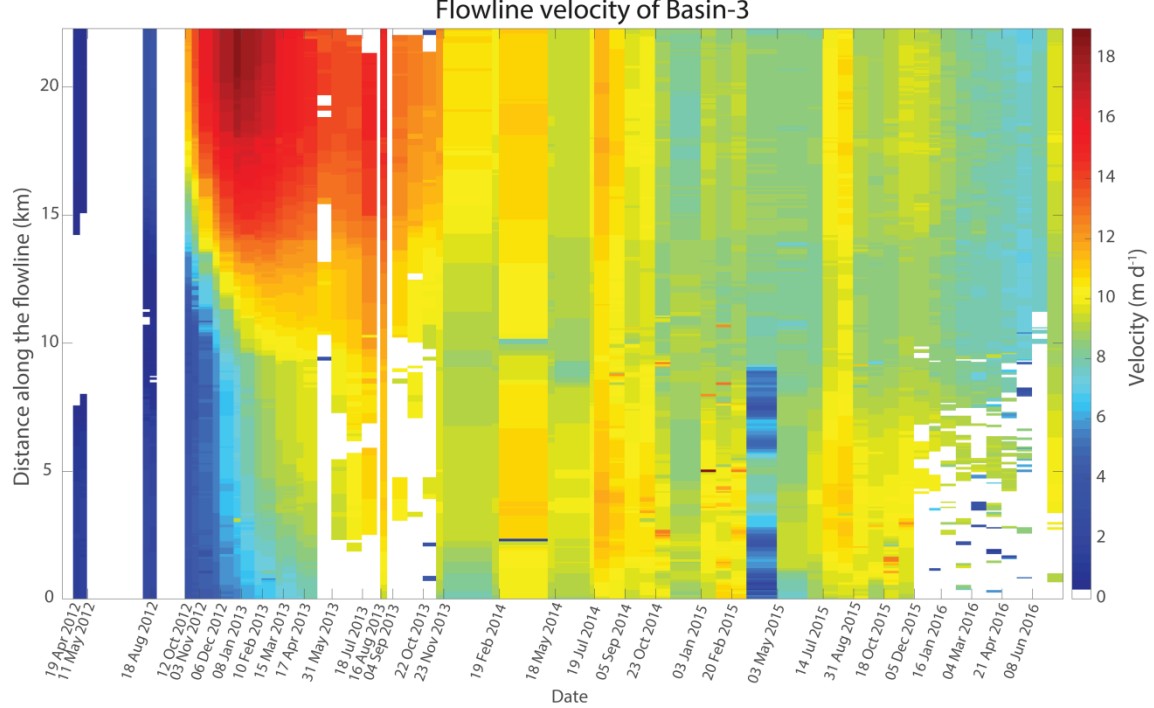

Figure 3: Surface velocity along the flowline profile of Basin-3 from the accumulation area (bottom) towards the calving front (top) between April 2012 and July 2016.





Figure 4: Velocity of IMAU GPSes on Basin-3 between April 2008 and July 2016 and cumulative positive degree days (CPDD).



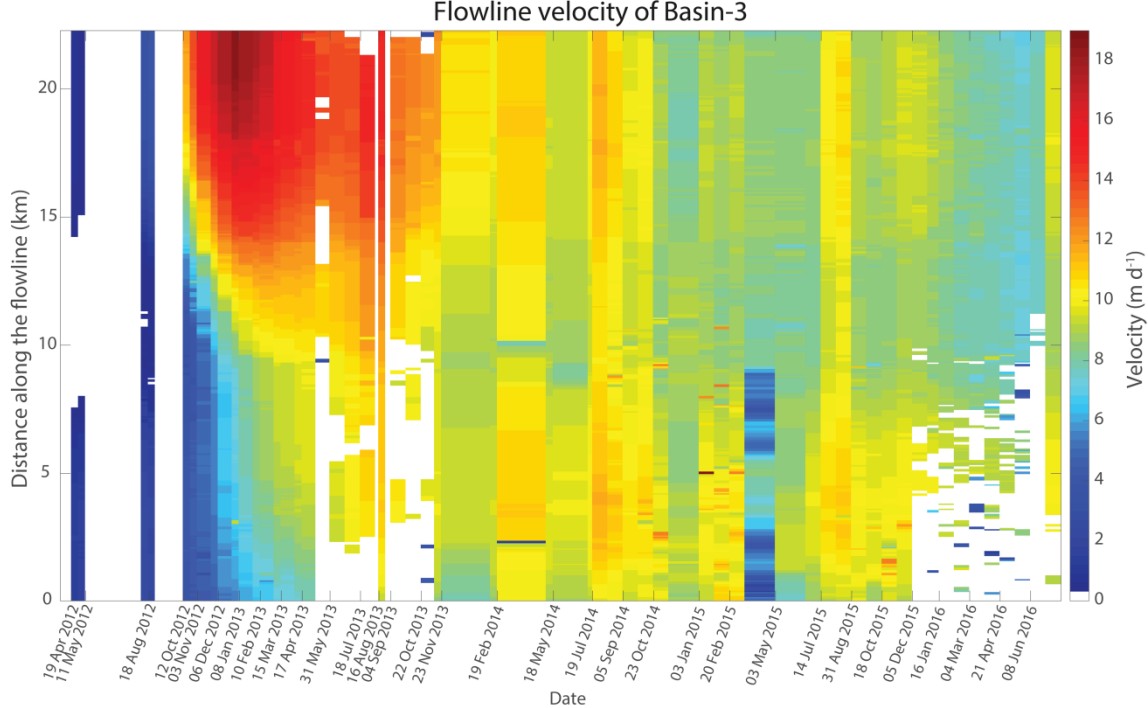

**Figure 5: Surface velocity profile along the fluxgate of Basin-3 from the south (bottom) to the north (top) between April 2012 and July 2016.**





**Figure 6: Surface velocity maps of Basin-3 and Basin-2: (a) TerraSAR-X: 30 April 2012 – 11 May 2012 (b) TerraSAR-X: 18 August 2012 – 29 August 2012 (c) TerraSAR-X: 12 October 2012 – 23 October 2012 (d) TerraSAR-X: 28 December 2012 – 8 January 2013 (e) TerraSAR-X: 16 August 2013 – 27 August 2013 (f) Radarsat-2 Fine: 19 July 2014 – 12 August 2014 (g) Radarsat-2**
5 **Fine: 14 July 2015 – 7 August 2015 (h) Radarsat-2 Wide Fine: 23 December 2015 – 16 January 2016 (i) Radarsat-2 Wide Fine: 2 July 2016 – 26 July 2016**



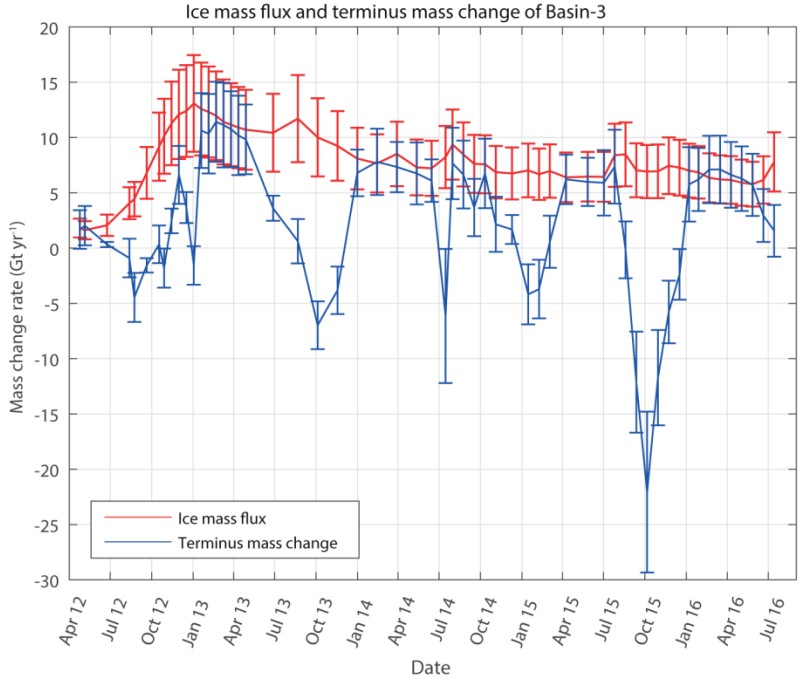

**Figure 7: Ice mass flux rate and terminus mass changes of Basin-3 between April 2012 and July 2016.**

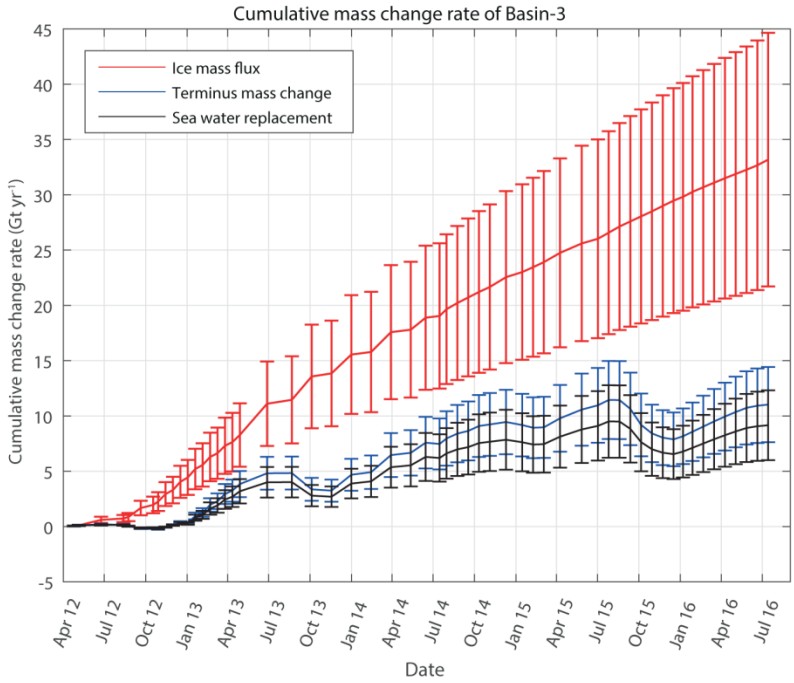

**Figure 8: Cumulative mass changes of Basin-3 between April 2012 and July 2016.**



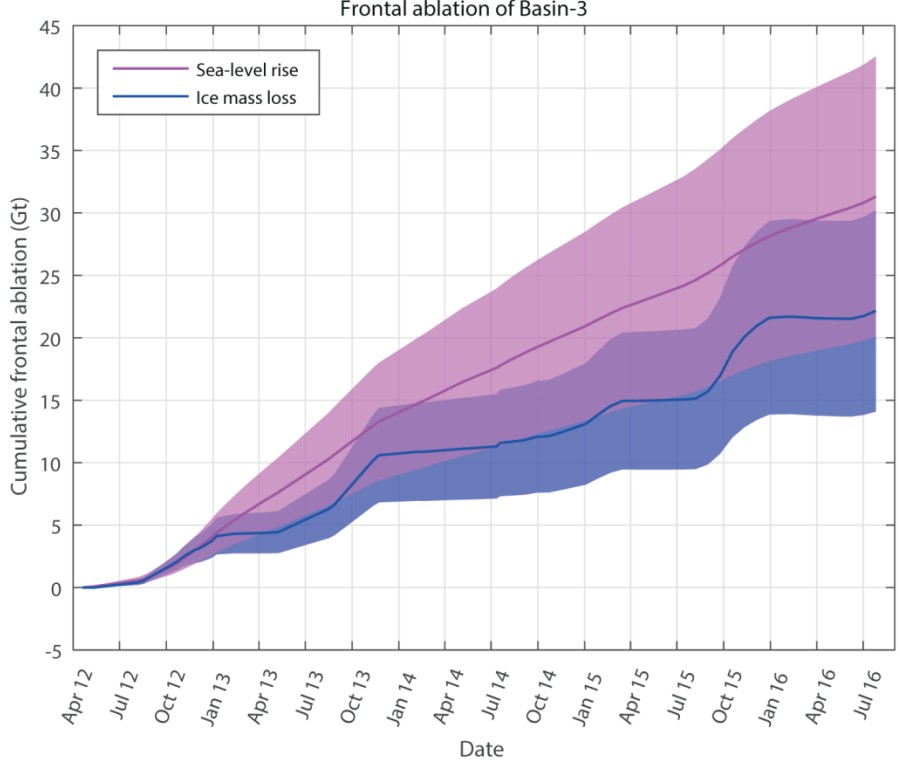

**Figure 9: Frontal ablation of Basin-3 from mass balance and sea level rise perspective between April 2012 and July 2016.**



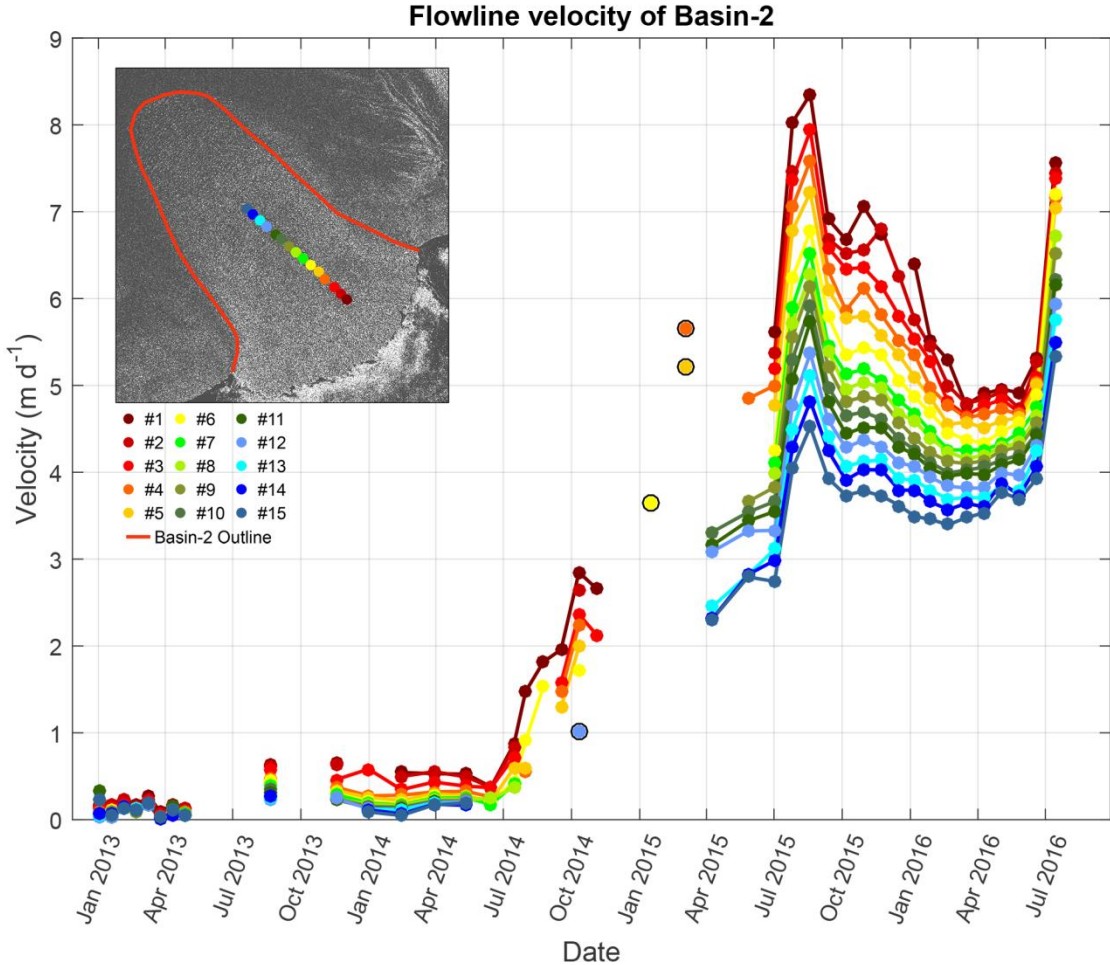

**Figure 10: Surface velocity at selected points along the flowline profile of Basin-2 between January 2013 and July 2016. Inset shows outline of the fast flowing northern Basin-2 with location of points.**




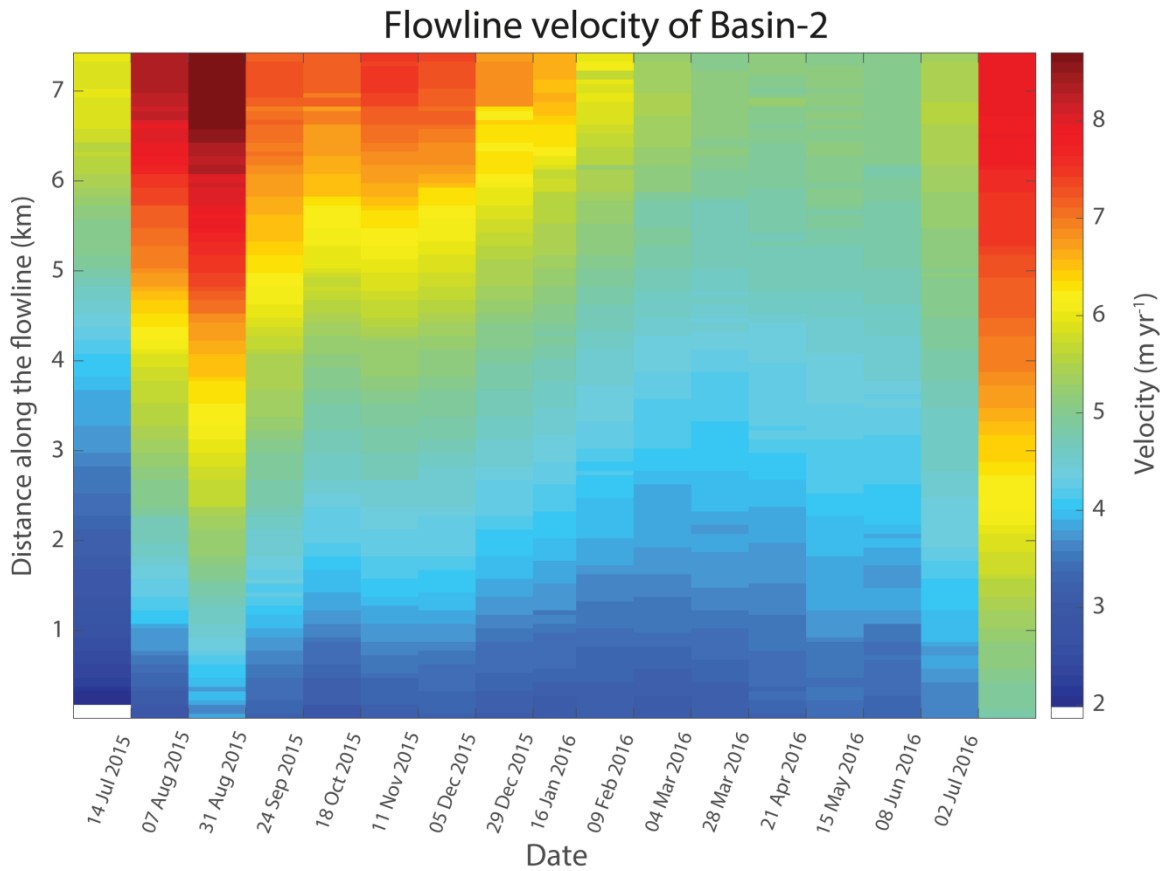

**Figure 11: Surface velocity along the flowline profile of northern Basin-2 from the accumulation area (bottom) towards the calving front (top) between June 2015 and July 2016**

5    **Table 1: Specifications and processing parameters of TerraSAR-X and Radarsat-2 data.**

| Sensor | TerraSAR-X | Radarsat-2 | Radarsat-2 | Radarsat-2 |
|---|---|---|---|---|
| Mode | StripMap | Fine | Wide Fine | Wide |
| Scene coverage (km) | 30 x 55 | 50 x 50 | 150 x 150 | 150 x 150 |
| Pixel resolution (m) | 1.7 x 2.0 | 7.8 x 4.9 | 9.6 x 5.2 | 24.1 x 5.2 |
| Matching window (pixel) | 300 x 344 (599 x 688) | 102 x 162 | 83 x 152 | 80 x 380 |
| Step size (pixel) | 50 x 57 | 6 x 10 | 10 x 19 | 8 x 38 |
| Resolution velocity map (m) | 100x100 | 50 x 50 | 100 x 100 | 200 x 200 |



**Table 2: Estimate of the total frontal ablation of Basin-3 and its components between 19 April 2012 and 26 July 2016.**

| Frontal ablation components | (Gt) | (Gt yr$^{-1}$) |
|---|---|---|
| Ice mass flux, $q_{fg}$ | 33.2 ± 11.5 | 7.8 ± 2.7 |
| Terminus mass change, $q_t$ | 11.0 ± 3.4 | 2.6 ± 0.8 |
| Terminus-seawater replacement, $q_{tsw}$ | 9.2 ± 3.2 | 2.1 ± 0.7 |
| Total frontal ablation | | |
| Mb perspective, $q_{mb} = q_{fg} - q_t$ | 22.2 ± 8.1 | 5.2 ± 1.9 |
| SLR perspective, $q_{sl} = q_{mb} + q_{tsw}$ | 31.3 ± 11.2 | 7.3 ± 2.6 |

10    **Table 3: Estimate of total frontal ablation of Basin-2 and its components between 20 June 2015 and 26 July 2016.**

| Frontal ablation components | (Gt) | | | (Gt yr$^{-1}$) | | |
|---|---|---|---|---|---|---|
| | min | | max | min | | max |
| Ice mass flux, $q_{fg}$ | 0.7 | 1.3 | 2.2 | 0.5 | 1.2 | 2.0 |
| Terminus mass change, $q_t$ | 0.3 | 0.4 | 0.6 | 0.2 | 0.4 | 0.6 |
| Terminus-seawater replacement, $q_{tsw}$ | 0.1 | 0.3 | 0.5 | 0.1 | 0.3 | 0.5 |
| Total frontal ablation | | | | | | |
| Mb perspective, $q_{mb} = q_{fg} - q_t$ | 0.3 | 0.8 | 1.6 | 0.3 | 0.7 | 1.4 |
| SLR perspective, $q_{sl} = q_{mb} + q_{tsw}$ | 0.5 | 1.1 | 2.1 | 0.4 | 1.0 | 1.9 |



**Appendix A**

**Table A1. TerraSAR-X (TSX) and Radarsat-2 (RS-2) acquisitions of Basin-3: repeat-pass period, start and end-date and maximum velocity of Basin-3. <sup>(*)</sup> The TSX velocity map of time step $t_{28}$ was scaled to 7 days for the calculation of the frontal ablation of Basin-3 to match with the start date of the RS-2 F data time series. <sup>(**)</sup> The RS-2 WF velocity map of time step $t_{48}$ was scaled to 18 days for the calculation of the frontal ablation of Basin-3 to match with the end date of the RS-2 F data time series.**

| ID of period | Sensor (Mode) | Repeat pass period (days) | Start- and end-date (yyyy mm dd) | Maximum velocity of Basin-3 (m d$^{-1}$) |
|:---:|:---:|:---:|:---:|:---:|
| $t_1$ | TSX | 11 | 2012 04 19 - 2012 04 30 | 3.2 |
| $t_2$ | TSX | 11 | 2012 04 30 - 2012 05 11 | 3.0 |
| $t_3$ | TSX | 88 | 2012 05 11 - 2012 08 07 | 4.0 |
| $t_4$ | TSX | 11 | 2012 08 07 - 2012 08 18 | 5.4 |
| $t_5$ | TSX | 11 | 2012 08 18 - 2012 08 29 | 5.6 |
| $t_6$ | TSX | 44 | 2012 08 29 - 2012 10 12 | 5.6 |
| $t_7$ | TSX | 11 | 2012 10 12 - 2012 10 23 | 12.5 |
| $t_8$ | TSX | 11 | 2012 10 23 - 2012 11 03 | 14.4 |
| $t_9$ | TSX | 22 | 2012 11 03 - 2012 11 25 | 16.5 |
| $t_{10}$ | TSX | 11 | 2012 11 25 - 2012 12 06 | 17.1 |
| $t_{11}$ | TSX | 22 | 2012 12 06 - 2012 12 28 | 17.8 |
| $t_{12}$ | TSX | 11 | 2012 12 28 - 2013 01 08 | 18.8 |
| $t_{13}$ | TSX | 22 | 2013 01 08 - 2013 01 30 | 18.1 |
| $t_{14}$ | TSX | 11 | 2013 01 30 - 2013 02 10 | 17.7 |
| $t_{15}$ | TSX | 22 | 2013 02 10 - 2013 03 04 | 17.0 |
| $t_{16}$ | TSX | 11 | 2013 03 04 - 2013 03 15 | 16.2 |
| $t_{17}$ | TSX | 22 | 2013 03 15 - 2013 04 06 | 15.5 |
| $t_{18}$ | TSX | 11 | 2013 04 06 - 2013 04 17 | 15.4 |
| $t_{19}$ | TSX | 22 | 2013 04 17 - 2013 05 09 | 14.7 |
| $t_{20}$ | TSX | 99 | 2013 05 09 - 2013 08 16 | 14.0 |
| $t_{21}$ | TSX | 11 | 2013 08 16 - 2013 08 27 | 15.8 |
| $t_{22}$ | TSX | 11 | 2013 08 27 - 2013 11 12 | 12.8 |
| $t_{23}$ | TSX | 11 | 2013 11 12 - 2013 11 23 | 12.5 |
| $t_{24}$ | TSX | 77 | 2013 11 23 - 2014 02 08 | 10.7 |
| $t_{25}$ | TSX | 11 | 2014 02 08 - 2014 02 19 | 10.6 |
| $t_{26}$ | TSX | 77 | 2014 02 19 - 2014 05 07 | 9.8 |
| $t_{27}$ | TSX | 11 | 2014 05 07 - 2014 05 18 | 9.9 |
| $t_{28}$ | TSX | 55 | 2014 05 18 - 2014 07 12 | 10.0 |
| $t_{29}$ | TSX | 11/7* | 2014 07 12 - 2014 07 23 | 10.5 |
| $t_{30}$ | RS-2 F | 24 | 2014 07 19 - 2014 08 12 | 12.3 |
| $t_{31}$ | RS-2 F | 24 | 2014 08 12 - 2014 09 05 | 11.4 |
| $t_{32}$ | RS-2 F | 24 | 2014 09 05 - 2014 09 29 | 10.8 |
| $t_{33}$ | RS-2 F | 24 | 2014 09 29 - 2014 10 23 | 11.0 |



| $t_{34}$ | RS-2 F | 24 | 2014 10 23 - 2014 11 16 | 10.1 |
| $t_{35}$ | RS-2 F | 48 | 2014 11 16 - 2015 01 03 | 9.2 |
| $t_{36}$ | RS-2 F | 24 | 2015 01 03 - 2015 01 27 | 10.6 |
| $t_{37}$ | RS-2 F | 24 | 2015 01 27 - 2015 02 20 | 10.2 |
| $t_{38}$ | RS-2 F | 24 | 2015 02 20 - 2015 03 16 | 10.6 |
| $t_{39}$ | RS-2 F | 48 | 2015 03 16 - 2015 05 03 | 9.1 |
| $t_{40}$ | RS-2 F | 48 | 2015 05 03 - 2015 06 20 | 9.5 |
| $t_{41}$ | RS-2 F | 24 | 2015 06 20 - 2015 07 14 | 9.9 |
| $t_{42}$ | RS-2 F | 24 | 2015 07 14 - 2015 08 07 | 10.9 |
| $t_{43}$ | RS-2 F | 24 | 2015 08 07 - 2015 08 31 | 11.2 |
| $t_{44}$ | RS-2 F | 24 | 2015 08 31 - 2015 09 24 | 10.4 |
| $t_{45}$ | RS-2 F | 24 | 2015 09 24 - 2015 10 18 | 10.2 |
| $t_{46}$ | RS-2 F | 24 | 2015 10 18 - 2015 11 11 | 10.2 |
| $t_{47}$ | RS-2 F | 24 | 2015 11 11 - 2015 12 05 | 10.4 |
| $t_{48}$ | RS-2 F | 24 | 2015 12 05 - 2015 12 29 | 10.2 |
| $t_{49}$ | RS-2 WF | 24/18** | 2015 12 23 - 2016 01 16 | 10.0 |
| $t_{50}$ | RS-2 WF | 24 | 2016 01 16 - 2016 02 09 | 9.7 |
| $t_{51}$ | RS-2 WF | 24 | 2016 02 09 - 2016 03 04 | 9.7 |
| $t_{52}$ | RS-2 WF | 24 | 2016 03 04 - 2016 03 28 | 9.8 |
| $t_{53}$ | RS-2 WF | 24 | 2016 03 28 - 2016 04 21 | 9.5 |
| $t_{54}$ | RS-2 WF | 24 | 2016 04 21 - 2016 05 15 | 9.3 |
| $t_{55}$ | RS-2 WF | 24 | 2016 05 15 - 2016 06 08 | 9.0 |
| $t_{56}$ | RS-2 WF | 24 | 2016 06 08 - 2016 07 02 | 8.9 |
| $t_{57}$ | RS-2 WF | 24 | 2016 07 02 - 2016 07 26 | 10.2 |

**Table A2: Frontal ablation of Basin-3: input variables – values, sources and uncertainties; partially from (Dunse et al., 2015).**

| Variable | Value/Source | Uncertainty | Explanation |
|---|---|---|---|
| $zs_{fg}$ | 40 m (constant) | ± 30 m | The chosen values allow for elevations from flotation height of 10 m as lower limit and mean DEM height of 67 m as upper limit. The DEM originates from prior to surge initiation. GPS data since 2008 indicates extensional flow, and hence, dynamic thinning. |
| $zb_{fg}$ | Local bedrock map values along fluxgate | ± 30 m | Twice the accuracy in ice thickness measurement of ± 15 m used to derive the bedrock map (Dunse et al., 2012), thereby accounting for uncertainties |





| | | | |
|---|---|---|---|
| | | | introduced by gridding of spatial inhomogeneous measurements. |
| $H_{fg}$ | $zs_{fg}$- $zb_{fg}$ | ± 42 m | RSS of errors in $zs_{fg}$ and $zb_{fg}$ |
| $zs_t$ | 30 m (constant) | ± 20 m | Allows for calving front heights down to flotation and significantly larger than typical front height of 30 m (Moholdt and Kääb, 2012). |
| $zb_t$ | - 87 m (constant) | ± 30 m | Value represents the mean bedrock elevation within the observed range in front position with an uncertainty consistent to the one of $zb_{fg}$ |
| $H_t$ | $zs_t$ - $zb_t$ | ± 36 m | RSS of errors in $zs_t$ and $zb_t$ |
| $v_{fg}$ | Local value from TSX and RS-2 velocity maps | ± 0.43 m d$^{-1}$ (0.48 for $t_3$, 0.51 for $t_6$, 0.45 for $t_{20}$, 0.74 for $t_{22}$) | Uncertainty based on standard deviation (SD) of TSX and GPS velocities, yielding 0.43 m d$^{-1}$; for the long repeat cycles $t_3$, $t_6$, $t_{20}$ and $t_{22}$, additional uncertainty is added based on a comparison of the available data and reconstructed velocity profile, resulting in a SD of 0.21, 0.26, 0.12 and 0.60 m d$^{-1}$. |
| $\Delta a_t$ | 2 m resolution TSX backscatter; 5 m resolution RS-2 backscatter | ± 8 m (4 pixel); ± 20 m (4 pixel) | Digitizing error of calving front position results in uncertainty of $\Delta A_t$, determined by the root sum square of the deviation from minimum and maximum extent of the terminus at times $t_{i(start)}$ and $t_{i(end)}$. |



**Table A3: Frontal ablation of Basin-2: input variables – values, sources and uncertainties.**

| Variable | Value/Source | Uncertainty | Explanation |
|---|---|---|---|
| $zs_{fg}$ | 40 m (constant) | ± 30 m | The chosen values allow for elevations from flotation height of 10 m as lower limit and mean DEM height of 67 m as upper limit. The DEM originates from prior to surge initiation. |
| $zb_{fg}$ | Local bedrock map values along fluxgate | ± 30 m | Twice the accuracy in ice thickness measurement of ± 15 m used to derive the bedrock map (Dunse et al., 2012), thereby accounting for uncertainties introduced by gridding of spatial inhomogeneous measurements. |
| $H_{fg}$ | $zs_{fg}$ - $zb_{fg}$ | ± 42 m | RSS of errors in $zs_{fg}$ and $zb_{fg}$ |
| $zs_t$ | 30 m (constant) | ± 20 m | Allows for calving front heights down to flotation and significantly larger than typical front height of 30 m (Moholdt and Kääb, 2012). |
| $zb_t$ | - 55 m (constant) | ± 30 m | Value represents the mean bedrock elevation within the observed range in front position with an uncertainty consistent to the one of $zb_{fg}$ |
| $H_t$ | $zs_t$ - $zb_t$ | ± 36 m | RSS of errors in $zs_t$ and $zb_t$ |
| $v_{fg}$ | Local value from RS-2 velocity maps | ± 0.29 m d$^{-1}$ | Uncertainty based on standard deviation (SD) of RS-2 F and GPS velocities, yielding 0.29 m d$^{-1}$ |
| $\Delta a_t$ | 5 m resolution RS-2 backscatter | ± 20 m (4 pixel) | Digitizing error of calving front position results in uncertainty of $\Delta A_t$, determined by the root sum square of the deviation from minimum and maximum extent of the terminus at times $t_{i(start)}$ and $t_{i(end)}$. |