# Peer review of "Multi-year surface velocities and sea-level rise contribution of the Basin-3 and Basin-2 surges, Austfonna, Svalbard"

_The Cryosphere, 2017_

## Referee Comment (RC1) · Anonymous Referee #1 · 13 Apr 2017

General overview This paper contains some useful data that expands on previous knowledge of the surge of Basin-2 on Austfonna, and provides new information concerning the surge of adjacent Basin-3 to the south. It therefore has good potential, but I found the final text difficult to follow and lacking a clear focus and a clear advance on what has previously been published (particularly in comparison to Dunse et al., 2015: http://www.the-cryosphere.net/9/197/2015/tc-9-197-2015.pdf). For example, several of the figures are identical or only minor updates to the previous paper, and there appear to be major errors in others (e.g., Fig. 5 is identical to Fig. 3). There are also major jumps in logic and conclusions made with little evidence to back them up; for example: - It is stated at the base of p9 that 'external factors can play an important role in the

initiation of such kind of surges. . .', but surge initiation isn't really discussed anywhere in the paper. This paper only presents patterns after the surge has started. - The statement of more similar summer velocity peaks in recent years on p8, L22, isn't backed up by Fig. 4 (including the fact that the 2016 record is truncated) - It's argued on p10 that seasonal velocity variations are 'mainly due to differences in the amount of meltwater reaching the bed', but no rigorous analysis has been undertaken to prove this. Cumulative positive degree days have been plotted in Fig. 4, but these aren't analyzed in a meaningful way and the connection to the magnitude of seasonal velocity variations isn't obvious.

Overall, the paper needs tightening up and needs to be more clearly differentiated from what's been published before. A better organized and more developed presentation and analysis of the results also needs to be undertaken to back up the discussion and conclusions.

Specific comments P1, L12: clarify over which period the 'terminus velocity slowed down by ∼50% until spring 2014' (e.g., in comparison to when the surge first started in 2012?) P1, L14: the sentence 'Summer speed-ups were superimposed even on the otherwise fast surge motion' doesn't particularly well describe the complex ice velocity variations in Fig. 4. P1, L17: units of Gt isn't a sea level rise equivalent, this is mass flux; units need to be in mm to be a sea level rise equivalent. It's also not clear how this sentence about 'Additional advance of the terminus. . .' relates to the previous sentence: e.g., do they both refer to the same period? It would perhaps be better if they were combined into one sentence. P1, L22: same issue as L17. Gt isn't the correct units for s.l. rise equivalent P1, L26: I would change this to 'most recent projections', since dynamics are now included in some projections P1, L27: define how big 'prominent speed-ups' are – e.g., increase by an order of magnitude? P1, L30: change 'bed changes' to 'bed which changes'. It would also be useful to add a sentence to explain why the hydrological switch occurs – i.e., connection to changes in overburden pressure

[Figure]

P2, L2: polythermal should not be hyphenated P2, L4: I don't believe that more heat generated at the bed was the only cause. The increase in ice thickness also results in better insulation of the bed and a reduced thermal gradient between the bed and surface, resulting in better retention of existing heat (e.g., geothermal) P2, L6: change 'show' to 'shows' P2, L8: the number of 345 surge-type glaciers doesn't mean much by itself. Need to also provide this as a % of the total glacier population in Svalbard P2, L11-13: could also be useful to introduce the findings of Sevestre and Benn (2015) here of the causes of glacier surging P2, L23: specify the distance of terminus advance, to enable comparison with the changes of Stonebreen described in the next sentence P2, L24: specify that Edgeøya is in SE Svalbard for those who may be unfamiliar with it P2, L28: change 'that also overarching external factors can play...' to 'that overarching external factors can also play...' P2, L32: 'novelty' is a strange word to use here. Something like 'focus' would be better.

P3, L8: clarify if 'length' is the centreline length? P3, L12: The sentence 'The ice flow changed rapidly over the past years' is awkward; wording would be better with something like 'The ice flow has changed rapidly in recent years' P3, L13: describe what the velocities were during the 'stepwise acceleration' (or refer to Fig. 4?) P4, L5: change 'between to displacement' to 'between displacement' P4, L8: provide the repeat orbit cycle of RS-2, so that the choice of pixel spacing is clear and for comparison with TSX P4, L12: provide resolution for DEM P4: most speckle tracking velocity papers do not use data from the summer due to the problems of melt reducing coherence between scenes. However, it seems that data from throughout the summer was used in this study, so a comment about whether loss of summer coherence was a concern would be useful in the methods section.

P5, L5-10: it seems that this is a very long quote from Dunse et al. (2015). It's unusual to have such a long quote in a scientific paper, so it would be better to rewrite this in your own words. P5, L13: clarify if the GPS vs. SAR-derived velocities were computed over exactly the same period? P5, L16: are there units for the standard deviation? (e.g.,

m d-1?) P5, L31: 'governing' doesn't seem to be the correct word here; something like 'dominates' would be better. Also provide a reference or more explanation to back up your assumption that surface velocity equals depth-averaged velocity, since most textbooks argue that surface velocity is greater.

P6, L3: change 'calculated to' to 'calculated to be' P6, L15: I assume that you're referring to air temperature here? Clarify since you also refer to snowpack temperature in the previous sentence. P6, L24 (and elsewhere): I don't think that 'GPSes' is a word'; change to 'GPS units'. Also change 'towards south-east' to 'towards the south-east' P6, L25-26: it would be useful to include a figure that shows the change in direction of the GPS units over time as they're drawn towards the fast flowing part of the glacier or pushed towards the margins (e.g., as inset in Fig. 4?) P6, L27-30: I'm unclear as to what is meant by 'the summer maximum overlies the surge'. Do you mean that a summer velocity maximum is superimposed on the high background surge velocities? The text describing the summer 2014 velocities on L29 is also difficult to follow: it's unclear as to what part of the velocity originates from the surge, and which part from the summer speed up. A clearer explanation of all the patterns described in this para would therefore be useful. P6, L27: Fig. 5 appears to be identical to Fig. 3; based on the caption I think that a different image was meant to be used for Fig. 5. P7, L2: a basic description of the glacier width and depth values used to calculate these values would be useful (the values in Tables A2 and A3 don't really provide specific values). E.g., 'Over a glacier terminus width of ?? km and average ice thickness of ?? m, we determined that total ice mass flux was…'. Also values for the average terminus advance in km.

P7, L5: similar to comments above, Gt isn't a unit of 'sea level rise equivalent'. P7, L10: I can find no reference to Fig. 8 or Fig. 9 in the text, but it seems that they should be referenced in this section (although they both seem to show similar things, so could perhaps be merged) P7, L13: reference is made to Fig. 6d here in relation to the speed up of Basin-2, but I don't see any evidence for speed up in this figure. Instead

the speed up is really only clear in Fig. 6g to 6h P7, L25-26: this sentence is poorly worded and doesn't really make sense (e.g., 'short enormous summer speed-ups'?!). It's also unclear where it's referring to as no figures show the velocities for the southern part of Basin-2. Either delete or reword and expand to provide a proper explanation of these velocity patterns and how they relate to those shown for the northern part of Basin-2. P7, section 4.2.2: similar to the comments for section 4.1.2 it would be useful to provide some basic physical characteristics for these measurements (e.g., average terminus width, ice thickness)

P8, L15-18: a figure that clearly shows the bedrock topography would be useful to back up this explanation: bedrock topography is shown in Fig. 1b (although this isn't referred to here), but there is insufficient detail in this figure compared to the patterns described in the text. A plot of the bedrock topography along the centreline shown in Fig. 3 could also help P8, L20: it's stated that 'summer maxima in. . . 2016 coinciding with the melt period', but no CPDD values are shown for 2016 to indicate when the melt period is P8, L22: the statement of more similar summer peaks in recent years isn't backed up by Fig. 4: only 2 GPS units were working in 2015, and in 2016 the velocity increase for the red GPS was much greater than the blue GPS (although the records are truncated) P8, L24: provide a quantification of how much higher the velocities were in autumn 2015 compared to other years P8, L29: the shear margin can't be seen in Fig. 1a, and this is labelled as a TerraSAR-X image in the figure caption rather than a RS-2 image. Fig. 1c also isn't a Landsat 8 image, and the shear margin isn't obvious (needs to be labelled). If reference is made to surface lowering here it should also be better incorporated into the rest of the discussion in this section, and in particular the velocity changes.

P9, L2: change 'results on' to 'results of' P9, L15-18: this text confidently describes what was happening at the glacier bed, but this is all conjecture and should therefore be described as such and backed up with evidence P9, L20: reference to Sentinel-1A data here seems to be irrelevant, since no successful results were found P9, L25:

description of the merging of the northern and southern branches of Basin-3 hasn't previously been made in the paper, so this explanation should be expanded upon (and reference made to Fig. 6a?) P9, 27: which bedrock map (Fig. 1b? – if so, refer to it!) P9, L33: I don't agree that you've demonstrated that 'external factors can play an important role in the initiation of such kind of surges'. You really haven't even spoken about factors that could have caused initiation in this paper, just the general patterns once surges have started. You demonstrate that external factors (i.e., summer melt) can modify velocity patterns once a surge has initiated, but not that they're a factor in the first place. Better reference to similar previous studies that talk about surge initiation is also needed, such as http://onlinelibrary.wiley.com/doi/10.1029/2002JB001906/full

P10, L1-2: this seems to be a throwaway comment, and I would delete it. You've demonstrated no connection to climate change in this paper, and have not even evaluated it as a potential influence. P10, L11: you haven't undertaken a rigorous analysis of whether the magnitude of seasonal velocity variations is related to the amount of meltwater reaching the glacier bed (CPDDs are just plotted at the base of Fig. 4, but no statistical analysis is undertaken of their influence). Based on your comments it seems that the stage of the surge is perhaps just as moment, or perhaps more important than CPDDs. P10, L16-19: besides being awkwardly worded, I don't agree with the statements here (as also outlined in the comment for P9, L33 above). What 'instability' are you referring to here? In this paper you also don't talk about surge initiation, only what happened after it.

Figures and tables Figure 1a: add label for Basin-2, to match that for Basin-3. There is a label for 'Crevasse formation' in the legend, but this appears to be blank; remove the label if this isn't being used. Fig. 1b: this should be labelled as bedrock elevation rather than bedrock depth Figure 1c: add distance markers to the flowlines so that the patterns shown in Fig. 3, etc., can be more easily understood. Use the same colours for the GPS units in all three parts of figure 1 (currently red in parts a and b, multi-coloured in part c) Figure 3 caption: refer to Fig. 1c here for the location of the

flowline Figure 4: add legend or inset map to indicate which GPS is which, and their distance from the glacier terminus. Also refer to Fig. 1 for their locations (although the current Fig. 1c doesn't show the location of the purple GPS - GPS #3?) Figure 5: this is identical to Figure 3! Figure 6: show outlines of Basin-2 and Basin-3 in this figure (or at least in part 6a) Figure 8/9: merge or delete one of these since they show similar things? Fig. 10: add a distance scale to the inset map and add the flowline (with distance markers) used in Fig. 11 Table A1: why are the TSX velocity values in black and RS-2 values in green?

---

## Short Comment (SC1) · 27 May 2017

Schellenberger et al. provide a detailed extension to the growing glacier velocity record of the Basin-3 and Basin-2 surge events Austfonna, Svalbard. The amount of data that has been synthesized for this paper is impressive and provides the observational data needed to identify the causal mechanisms resulting in several large coincident surges in the area. Figures 4 and 6 that provide a detailed temporal record of ice flow are particularly impressive. I do however agree with Reviewer 1 that the study struggles to provide new insights and falls short of linking observed changes in velocity to changes in the force balance of the glacier. The paper already provides a strong observational record of extraordinary changes in glacier behavior. All that is needed is a cleaning

up the text, removing many of the velocity and mass change numbers from the text that are better dealt with in the figures and table, and an enhanced discussion of likely mechanisms driving the observed changes, particularly with Phase-4 of the glacier surge, which is uniquely observed in this manuscript.

Specific Comments: Title – The title should focus on the interesting dynamics of the surge and not the sea-level contribution, which is negligible

P1L15 – precision with which mass change is being reported is not supported but the errors, I would recommend rounding to the nearest Gt.

P1L23 – instead of sea-level rise equivalent you could say added x Gt/yr. to ocean mass and displaced another y Gt/yr. SLE units are in units of height and are not appropriate for such a small contribution.

P2L19 – I'm not convince cryo-hydrolic warming plays a role here… from Figure 4 it doesn't seam like there is any strong relationship between speedup and DPDD.

P2L38 – "that also overarching external factors can play an important role" to "that overarching external factors can also play an important role"

P3L17 – "at the" to "at their"

P3L25 - "prolonged" to "extended"

P4L13 – delete "using Matlab"

P4L31 – did you define IDW? Maybe I missed it.

P5L4 - "prolonged" to "extended"

P5L12toL19 – The error should be characterized by RMSE, not linear fits and the standard deviation.

P6L24 – should this be Figure 4 not Figure 3?

P7L4 – incorrect propagation of errors 33.2 +/- 11.5 minus 11.0 +/- 3.4 = 22.2 +/-

sqrt(11.5ˆ2 +3.4ˆ2) = 12. Check throughout manuscript.

P7L13 – delete "neighboring"

P7L20 – delete "up to"

P7L25 – "also short enormous summer" to "also showed large summer"

P7L30 – "July 2016 to . . ." to "July 2016 to be"

P8L24 – "this had also influence" to "This also had an influence on"

P8L29 – Figure 1c shows a radar image not a Landsat 8 image

P9L12 – "but velocity increased instead steadily" to "but velocity instead increased steadily"

P9L33 – "indicate thus that also external" to "indicate that external"

P10L2 – "high arctic or" to "high Arctic, or"

P10L17 – delete "also here"

P10L18 – "mechanism to summer melt which triggered the Basin-3 surge plays an important role." to "mechanism to summer melt, which triggered the Basin-3 surge, plays an important role here as well."

P10L20 – "surge to still last for a few years." to "surge to last for a few years longer."

P10L22&23 – delete "even"

P10L24 – delete "especially.

Figure 2: RMSE is the most relevant statistic

Figure 3 – accurately define "distance along flowline".

Figure 5 – wrong figure here. . .

Figure 6 – I would suggest including the basin boundaries shown in Figure 1.. would make it easier to flow between panels

Figure 10 & 11- great figures

[Figure]

---

## Referee Comment (RC3) · A. Luckman (Referee) · 1 Jun 2017

Having considered the reviews by Anonymous Reviewer 1 and Alex Gardner, I don't want to pile on too much more misery, because in the end this work presents a great deal of interesting data about the Austfonna surges, and has the potential to provide a useful update to the nature of these events. But a few further things need to be mentioned.

GENERAL COMMENTS

In accordance with the other reviews, this paper seems to be rather sloppy in places

both in the way it is written, and in attention to detail. I hope that more careful internal review by co-authors during revisions will solve some of these problems. I will refrain from pointing out every small issue, and I look forward to reading an updated manuscript. However, some important issues seem to be recurrent and some not yet mentioned, and these are dealt with below.

SPECIFIC COMMENTS

Several strong assertions are made about the location and volume of meltwater, and the nature of mechanisms occurring at the bed (e.g. p7, line19; p8, line19; p10, line11; several other places). No observations have been made of meltwater or the bed so it is important to clearly separate what is observed and what is inferred from the observations you have made.

The strongest claims of this paper are made about the frontal ablation rates and sea-level contribution, yet important aspects of these calculations and data are not presented or discussed. The reason that I am troubled by this is that you seem to reach very different conclusions to those we reached in Luckman et al. (Nature Comms., 2015) in which the frontal ablation rates did not change significantly between surge and pre-surge conditions of Aavatsmarkbreen (admittedly a much smaller glacier). Firstly (and I don't say this simply to gain a new citation) it is a bit of an omission not to have discussed your results in comparison to our paper, and I encourage you to check our reference list to be sure that there are no other comparisons to frontal ablation rate papers you have over-looked. Moving on, I am not yet convinced by your method because you give very little detail about the geometry of the glacier and how it is changing and in my experience, ice-front change normally dominates the calculated ablation rate. I would like to see the series of ice-front positions from which you calculate volume loss and, much more importantly, you need to discuss the potential impact of changing surface topography on the values you calculate. How can you justify constant thickness values in Page 6 line 5 (even though you do mention potential errors) during a surge? If so much ice is being lost, where is it going? Do you see large or small calved

icebergs? How does the lost mass interact with sea-ice? What proportion of loss is through ocean-melt? It is plausible that, because Basin 3 is in a different setting to Aavatsmarkbreen, and the ice-front is diverging into deeper water, your frontal ablation values are reasonable, but you really need to present much more information, and discuss all of the confounding factors, to allow the reader to be able to understand and agree with your conclusions. This is a big claim and needs clearer evidence to support it.

Several seemingly novel compound terms are used as if they are well-accepted: e.g. "frontal ice plug", "sheer-tearing of the ice-sediment interface", "hydro-thermodynamic". In my view these terms obscure rather than clarify the discussions and it would be better to explain things more descriptively, and without the introduction of new jargon.

Page 4, line 21: The use of 'time-steps' does not seem to be helpful, is not consistent with the rest of the text, and, if this really is the best way to explain things, they need to be introduced before they are referred to.

The final paragraph of the conclusion is all about Sentinel-1, yet the paper does not use these data so I find this a strange and unhelpful way to end.

In summary, it will be great to see these observations published and I look forward to seeing the paper revisions, but the frontal ablation calculations (because they seem to be the result you are promoting) need to be presented more completely and more thoughtfully, and more attention to detail needs to be paid elsewhere.